# Antenatal Fear of Childbirth as a Risk Factor for a Bad Childbirth Experience

**DOI:** 10.3390/healthcare11030297

**Published:** 2023-01-18

**Authors:** Azahara Rúger-Navarrete, Juana María Vázquez-Lara, Irene Antúnez-Calvente, Luciano Rodríguez-Díaz, Francisco Javier Riesco-González, Rocío Palomo-Gómez, Juan Gómez-Salgado, Francisco Javier Fernández-Carrasco

**Affiliations:** 1Department of Surgery, Hospiten Estepona, 29689 Estepona, Spain; 2Nursing Department, Faculty of Health Sciences of Ceuta, University of Granada, 51001 Ceuta, Spain; 3Department of Obstetrics, Hospital Universitario Punta de Europa, 11207 Algeciras, Spain; 4Department of Obstetrics, La Linea de la Concepción Hospital, 11300 La Línea de la Concepción, Spain; 5Department of Sociology, Social Work and Public Health, Faculty of Labour Sciences, University of Huelva, 21007 Huelva, Spain; 6Safety and Health Postgraduate Programme, University of Espíritu Santo, Guayaquil 092301, Ecuador; 7Department of Nursing and Physiotherapy, Faculty of Nursing, University of Cádiz, 11207 Algeciras, Spain

**Keywords:** fear of childbirth, childbirth experience, obstetrics, pregnancy, childbirth

## Abstract

Giving birth is one of the most impressive experiences in life. However, many pregnant women suffer from fear of childbirth (FOC) and experience labour in very different ways, depending on their personality, previous life experiences, pregnancy, and birth circumstances. The aim of this study was to analyse how fear of childbirth affects the childbirth experience and to assess the related consequences. For this, a descriptive cross-sectional study was carried out in a sample of 414 women between 1 July 2021 and 30 June 2022. The Birth Anticipation Scale (BAS) was used to measure fear of childbirth and the Childbirth Experience Questionnaire (CEQ-E) was applied to measure satisfaction with the childbirth experience. Fear of childbirth negatively and significantly predicted the childbirth experience. In addition, women who were more fearful of childbirth were found to have worse obstetric outcomes and a higher likelihood of having a caesarean delivery (*p* = 0.008 C. I 95%). Fear behaved as a risk factor for the birth experience, so the greater the fear, the higher the risk of having a worse birth experience (OR 1.1). Encouraging active listening and support strategies may increase pregnant women’s confidence, thus decreasing their fear of the process and improving their childbirth experience.

## 1. Introduction

Childbirth is a decisive, meaningful, and life-changing experience. In this sense, a joyful and positive experience is expected, but fear related to childbirth (or some kind of fear) cannot be ruled out as it is difficult to predict how the process will unfold. Thus, fear of childbirth (FOC) can fluctuate during pregnancy, with up to 80% of women experiencing some form of childbirth-related fear at some point, which becomes more intense in the last weeks of pregnancy [1,2].

FOC is a problem that occurs among nulliparous, primiparous, and multiparous women, and that has health consequences and implications for labour and the puerperium [1,2]. O’Connell et al. found that more than 40% of new mothers and more than 30% of multiparous women had a high level of fear of childbirth [1].

FOC exists on a spectrum from low to high. Its extreme expression is phobic fear, which involves avoidance behaviours that typify a phobic condition. Women with an extreme fear of childbirth may be scrupulous about contraception or resort to voluntary termination of pregnancy to avoid having to face childbirth at all costs. This phobic fear is known as tokophobia, a severe fear of pregnancy and childbirth [3,4]. However, even when FOC is not a phobic disorder, it can encompass a wide range of emotions during pregnancy, such as anxiety and stress, which also reflect a spectrum of maternal distress [5,6]. In addition, FOC can cause significant problems during labour which may translate into increased pain and a prolongation of the first and second stages of labour, together with a strong sense of dissatisfaction [7,8,9]. Furthermore, fear has a stronger association with pain and duration of labour than stress [4]. Epidural anaesthesia initially reduces pain, but interestingly, women who request it are more fearful [10]. This problem is also found in 7–22% of elective caesarean sections chosen by the mother without a medical justification, which increases the caesarean section rate [11]. Ryding et al. also found that severe fear can cause labour to result in an emergency caesarean section [12].

Possible causes of FOC are related to several factors, such as women’s personal, internal, and external conditions, i.e., mental health problems (such as anxiety disorders) and previous experiences of trauma and abuse. Additionally, social circumstances, such as poor social support, unemployment, and economic problems, have an effect on the likelihood of developing FOC [13,14]. To this, cultural beliefs about childbirth as a risky medical event are added, as they may also influence the development of FOC [15,16].

Women’s experiences of pregnancy, labour, and childbirth are multidimensional [17] and can include everything from joy and satisfaction to anxiety and horror. Some pregnant women are gripped by negative feelings and may develop a fear of childbirth, which can have consequences for their wellbeing and health [18].

Fear of childbirth experiences in women appear to be related to their emotional wellbeing, stress symptoms, impacts on daily life, and desire for a caesarean section at their next birth [13]. Women who fear childbirth may feel a lack of confidence in childbirth, be influenced by negative stories about childbirth, fear labour pain or loss of control, and fear physical injury during childbirth. Women who give birth after a previous negative birth experience often fear another bad birth experience [19].

The emotional and psychological wellbeing of women significantly contributes to their perceptions and experiences of pregnancy and childbirth. Poor emotional health is associated with increased fear of childbirth and risk of depression [20], birth trauma [21,22,23,24], inability to interact positively with the baby and meet the child’s developmental needs [25,26], and can also act as a stressor for the couple’s relationship [26,27].

The aim of this research was to analyse how fear of childbirth (FOC) affects the childbirth experience and to assess how it influences the other obstetric and socio-demographic variables.

## 2. Materials and Methods

### 2.1. Study Design

A descriptive cross-sectional study was designed using questionnaires.

The study variables were ‘fear of childbirth’ (FOC) using the Birth Anticipate Scale (BAS) score [28] and ‘childbirth experience’ using the Childbirth Experience Questionnaire in its validated Spanish version (CEQ-E) [29].

In addition, sociodemographic variables were taken into account: age, marital status, educational level, and income. Furthermore, obstetric variables were included, such as having attended antenatal classes, mode of onset of labour, analgesia received during delivery, type of delivery, and receiving stitches after delivery [30].

### 2.2. Population and Sample

The study population consisted of a total of 3269 pregnant women under follow-up and who gave birth in two public hospitals of Andalusia: Hospital Punta de Europa in Algeciras (Cadiz) and Hospital Costa del Sol in Marbella (Malaga), between July 2021 and June 2022.

Establishing a confidence level of 95% and with a maximum error of 5%, the optimal sample should consist of at least 344 subjects. Finally, a total of 414 subjects were recruited to allow for possible losses, estimated at around 20%.

The sample selection process was carried out systematically, randomly selecting women who visited any of the hospitals for a follow-up of their pregnancy, who met the inclusion criteria and wished to voluntarily participate in the study.

The inclusion criteria were women who attended the pregnancy follow-up consultation in the 35th week of amenorrhoea and who subsequently gave birth in any of the two study hospitals by eutocic, dystocic, and/or caesarean delivery without peripartum pathologies, such as placental abruption, prolapsed cord, or acute foetal distress. In addition, the newborns must have been born healthy and without any neonatal pathology and should have not required more time than usual in hospital.

The exclusion criterion was the existence of a language barrier (patient could not write, read, or even speak in Spanish). In many cases, the companion or direct relative did understand Spanish, but even so, this requirement was maintained as an exclusion criterion.

### 2.3. Instruments

Two validated questionnaires were used for data collection: the Birth Anticipation Scale (BAS) [28] and the Childbirth Experience Questionnaire in its validated Spanish version (CEQ-E) [29]. In addition to these two tools, sociodemographic data were collected from the patients.

The BAS includes a six-item scale to measure fear of birth. Participants express whether they feel nervous, worried, fearful, relaxed, terrified, or calm regarding the birth of the baby, using ‘extremely’, ‘quite a lot’, ‘moderately’, ‘a little’, and ‘not at all’ as response options. The total score is the sum of the responses; the higher the score, the more fearful the woman is of the upcoming birth. The total score can range from 6 (no fear) to 30 (extreme fear) and the overall Cronbach’s Alpha obtained for this scale was 0.82. Scores are classified into three categories: between 6 and 13—‘low fear’; between 14 and 20—‘intermediate fear’; and between 21 and 30—‘high fear’ [28].

Given that this questionnaire was validated in the United States by Elvander et al. [28], in order to use it in the present study environment and specific population, the corresponding cross-cultural adaptation was carried out. The reliability analysis using Cronbach’s alpha for this resulting version was slightly higher than that of the original test, with a value of 0.87.

The CEQ-E includes 22 items referring to the childbirth experience. Responses to the first 19 items are scored on a 4-point Likert scale (1. strongly agree; 2. mainly agree; 3. mainly disagree; 4. strongly disagree) and the last three items are assessed using a visual analogue scale, i.e., pain memory, perceived safety, and control.

The CEQ-E questionnaire consists of 4 domains: ‘own capacity’ (8 items related to sense of control, personal feelings, and labour pain); ‘professional support’ (5 items about professional care); ‘perceived safety’ (6 items regarding sense of security and memories from the childbirth); and ‘participation’ (3 items related to own possibilities to influence position, movements, and pain relief during labour and birth). The internal consistency reliability of the CEQ-E was good for the overall scale (0.88) and for all the subscales (0.80, 0.90, 0.76, 0.68 for ‘own capacity’, ‘professional support’, ‘perceived safety’, and ‘participation’, respectively), similar values to the ones in the original version [18].

### 2.4. Data Collection

Data were collected from 1 July 2021 to 30 June 2022. Data were obtained from the retrospective patient management database of the Andalusian Public Health System (Spain).

Once the patient was selected, a member of the research team contacted her by telephone to inform her of the characteristics of the study and invite her to participate. When the patient gave her consent, she was sent a Google forms link via WhatsApp so that she could access the virtual questionnaire conveniently and securely from her own phone.

### 2.5. Data Analysis

The descriptive analysis of quantitative variables was performed using measures of central tendency and dispersion. Categorical variables were described in absolute numbers and percentages. In the bivariate statistical analysis, to compare quantitative variables and after performing the Kolmogorov–Smirnov normality test, non-parametric tests were used: Mann–Whitney U test and Kruskal–Wallis test. Finally, a multivariate binary logistic regression model was developed to establish associations between the different independent variables and the dependent variable (childbirth experience). Confidence intervals (CI) were obtained at 95% and a significance level of *p* < 0.05. The statistical study was carried out using the IBM SPSS Statistics version 26.

For the validation of the BAS questionnaire, the original questionnaire was translated into Spanish by two native translators. It was then evaluated by a group of experts to check whether the items were adapted to the target context. The questionnaire was then pilot-tested with a group of 30 women to confirm the correct understanding of the questions. The next step was to administer the questionnaire to the study sample and to carry out a subsequent factor analysis of all its components. Finally, this factor analysis was validated using the AMOS programme, confirming the construct validity.

### 2.6. Ethical Aspects

The general principles of the Declaration of Helsinki, updated in 2013 in Fortaleza (Brazil), were considered throughout this study by the entire research team. In addition, the provisions of current Spanish legislation on biomedical research (Law 14/2007, of 3 July, on Biomedical Research) and Law 41/2002, of 14 November, on patient autonomy and rights and obligations regarding clinical information and documentation were followed. All personal data were protected in accordance with Organic Law 15/1999, of 13 December, on the Protection of Personal Data. Permission to conduct this study was obtained from the Andalusian Biomedical Research Ethics Committee (code 47-N-20).

## 3. Results

### 3.1. Validation of the BAS Questionnaire

A factor analysis was carried out for the validation of the BAS questionnaire.

The results of the validation were as follows: The Kaiser–Meyer–Olkin measure of sampling adequacy was 0.84 and Bartlett’s test of sphericity had a significance of 0.001. As in the original questionnaire, only a single domain was obtained which explained 62% of the variance. When applying the test–retest reliability analysis using Crombach’s alpha, a value of 0.87 was obtained. This value was somewhat higher than that of the original scale.

### 3.2. Descriptive Analysis

The mean age of the women was 32 years, with a minimum age of 16 years and a maximum age of 44 years. The median value for the Birth Anticipation Scale (BAS) was 16, with a SD of 4.84; the median value for the Childbirth Experience Scale (CEQ-E) was 64, with a SD of 11.27.

The descriptive analysis of the qualitative variables is shown in Table 1.

### 3.3. Correlational Analysis

Age was positively correlated with number of children (*p* = 0.001), and negatively with childbirth experience (*p* = 0.001). Number of children was positively and significantly correlated with childbirth experience, so the higher the number of children, the better the experience (*p* = 0.008).

Fear of childbirth was negatively correlated with the childbirth experience, so the more fearful the woman was, the worse the experience (*p* = 0.001). Although the correlations were significant on some occasions, they were generally very low, i.e., below 0.6 (Table 2).

### 3.4. Comparisons of Medians (—) and Bivariate Analysis

#### 3.4.1. Fear of Childbirth (BAS) and Childbirth Experience (CEQ-E) Related to the Different Sociodemographic Variables

Statistically significant differences were found between the different groups when relating fear of childbirth to level of education (*p* = 0.005) and income (*p* = 0.002). Similarly, these differences were also found when relating childbirth experience to marital status (*p* = 0.04) and income (*p* = 0.001). However, no statistically significant differences were found regarding the other associations (Table 3).

#### 3.4.2. Fear of Childbirth (BAS) and Childbirth Experience (CEQ-E) Related to Obstetric Factors

Statistically significant differences were found when relating childbirth experience (CEQ-E) to having had previous miscarriages. Thus, women who had had miscarriages had a better experience (*p* = 0.001). No such differences were found when relating this variable to fear of childbirth (BAS) (Table 4). Relevant differences were also found when relating the variable ‘attendance to antenatal classes’ to fear of childbirth (BAS), so women who attended these courses were less fearful of childbirth (*p* = 0.02). However, no differences were found when relating this variable to childbirth experience (CEQ-E). When analysing the influence that being accompanied during dilation and/or delivery may have on the outcomes, statistically significant differences were observed in relation to fear of childbirth (BAS) (*p* = 0.001) and childbirth experience (CEQ-E). Thus, accompanied women had a better childbirth experience (*p* = 0.001) (Table 4).

No statistically significant differences were found when relating the mode of onset of labour to the ‘fear of childbirth’ variable (BAS). However, significant differences were found when relating this variable to childbirth experience (CEQ-E), with women who started labour spontaneously having a better experience (*p* = 0.001) (Table 5).

The post hoc test showed statistically significant differences between the group in which labour had begun spontaneously and the group that had a planned caesarean section (*p* = 0.001).

Statistically significant differences were also found when relating the type of analgesia used during labour to fear of childbirth (BAS). Women who did not request any type of analgesia had the lowest levels of fear of childbirth (*p* = 0.001) (Table 5). The post hoc test for the ‘fear of childbirth’ variable (BAS) did not establish statistically significant differences between the different groups. Regarding the childbirth experience according to the type of analgesia chosen to alleviate labour pain, it could be observed that women who did not choose any type of analgesia were those who had a better childbirth experience (CEQ-E). On the other hand, the post hoc test for the ‘childbirth experience’ variable (CEQ-E) established that significant differences were found between the group of women who did not use any analgesia and the group of women who used opioids as a method of pain relief in labour (*p* = 0.001); between the group of women who chose opioids for labour pain relief and the group of women who chose epidural analgesia (*p* = 0.02); and between the group of women who did not use any analgesia for labour pain and the group of women who chose epidural analgesia (*p* = 0.001).

When relating the type of delivery to the ‘fear of childbirth’ scale (BAS), statistically significant differences were found. Women who had a eutocic birth were the least fearful (Table 5). Yet, the post hoc test for the ‘fear of childbirth’ variable did not establish statistically significant differences between the different groups.

Statistically significant differences were also observed for the association between the experience of childbirth and the type of delivery, so women who had a eutocic birth were the ones who obtained scores indicating a better childbirth experience. The post hoc test established that statistically significant differences were found between the group of women who had a eutocic birth and the group of women who had a caesarean (*p* = 0.001); and between the group of women who had a eutocic birth and the group who had an instrumental birth (*p* = 0.001).

No statistically significant differences were found when relating the ‘stitches received at delivery’ variable to the fear of childbirth scale (BAS). However, statistically significant differences were found between the different groups when relating the childbirth experience to the ‘stitches received at delivery’ variable, with women who had had no stitches at birth having the best childbirth experience. The post hoc test also showed statistically significant differences between the following groups: between the group of women who ended labour with no stitches and the group of women who ended labour with an episiotomy (*p* = 0.001); between the group of women who ended labour without stitches and the group of women who had a caesarean section (*p* = 0.001); between the group of women who ended the birth with stitches and the group of women who had a caesarean section (*p* = 0.001); and between the group of women who ended the birth with an episiotomy and the group of women who had a caesarean section (*p* = 0.001).

### 3.5. Multivariate Analysis (—) and Binary Logistic Regression Analysis

The multivariate analysis was performed with the ‘childbirth experience’ recoded variable (CEQ-E test score), with values of 1 (values above the mean) and 0 (values below the mean). The independent variables included were age, number of children, fear of childbirth (BAS test score), as well as those categorical variables that were statistically significant in the bivariate analysis: marital status, economic income, previous miscarriages, mode of onset of labour, analgesia used in labour, accompaniment during dilation and/or delivery, type of delivery, and stitches received after labour (Hosmer and Lemeshow Chi-square Test 16. 12 (*p* = 0.04); Nagelkerke’s R2 (0.47)) (Table 6).

Regarding age, it behaved as a risk factor for the ‘childbirth experience’ variable in such a way that the older the woman, the worse the childbirth experience.

Women with the highest income (over EUR 3000 per month) were 3.53 times more likely to have a better experience than those with the lowest income (less than EUR 1000 per month). Similarly, middle-income women (income between EUR 1000 and 2000 per month) were 2.4 times more likely to have a better childbirth experience than those with lower incomes (Table 6). Additionally, women who had had previous miscarriages were 1.59 times more likely to have a better childbirth experience than those who had had no previous miscarriages. Finally, women who were accompanied during dilation and/or delivery were 3.68 times more likely to have a better childbirth experience than those who were unaccompanied (Table 6).

Dependent variable: childbirth experience (0/1); independent variables: number of children, marital status, income, previous miscarriages, mode of onset of labour, analgesia, accompaniment during dilation and/or delivery, type of delivery, stitches received, fear of childbirth (FBS test score).

## 4. Discussion

Many pregnant women suffer from severe fear of childbirth (FOC) and experience this in very different ways, depending on their personality, previous life and birth experiences, pregnancy, and birth circumstances [1,9]. This variability highlights the interest in approaching this experience from different points of view, so that different perspectives can be provided. In this sense, the present study may offer relevant information to be considered in the process.

Among the main findings of the present study, women who were more fearful of childbirth were found to have a worse childbirth experience. It has been consistently published in the scientific literature that a higher FOC has been associated with unfavourable birth outcomes and worse subjective childbirth experiences [9].

Older women were more fearful of childbirth than younger women and, though this seems logical, the result can be surprising. It can be due to the fact that, as a general rule, age is a risk factor for complications during pregnancy [31,32]. In addition, older women are more likely to have had previous pregnancies, and therefore to have experienced childbirth before. If this previous experience was not the best, it can be expected that they will be more fearful of the new upcoming birth [33,34].

Women who attended antenatal classes had lower levels of FOC and, as a consequence, a better childbirth experience, most likely due to the information received in these courses. These findings are consistent with other recently published studies [30,31,32,33,34,35,36]. However, the fact that women who attended antenatal classes had a better experience of childbirth has not been statistically proven as, although they obtained higher childbirth experience scores (CEQ-E), the differences identified were not statistically significant.

The way in which labour was initiated was also a determinant as regards the level of fear of childbirth. Thus, women who had a planned caesarean section were the most fearful. This was probably due to the fact that vaginal birth is understood as a more natural process, whereas delivery by caesarean section implies a surgical intervention, with the risks that this may entail. In fact, women who were scheduled for induction of labour were also more fearful than those who had a normal delivery. These results are consistent with other studies, such as the one by Sydsjö et al. [37].

In Spain, epidural analgesia is the most commonly used method to relieve pain during labour [38]. FOC has been shown to negatively influence women’s pain during labour [7], and the present study results are in line with these statements as women who had higher levels of FOC were mostly those who requested this type of analgesia. This may be due to the fact that women with more fear have decreased tolerance to pain and more anxiety during the labour process, so the fact of enduring less pain reassures them. However, it was also observed that women who requested epidural analgesia had a worse childbirth experience than those who did not. Studies have shown that appropriate pain management during labour makes the childbirth experience more satisfying [39]. However, the experience depends on many factors, such as the length of the process, the difficulty of the second stage of labour, the way the woman is treated by the health personnel, etc., and not only on pain management [36]. In this sense, it is necessary to distinguish between the way in which the delivery process began and how it ended. A woman may have gone into labour spontaneously or even induced, but may eventually end up having a eutocic birth, an instrumental birth, or caesarean section. In the present study, women with higher levels of FOC had a caesarean section while those with lower levels of FOC had a eutocic birth. All in all, it is interesting to note these findings, which are consistent with the ones found in the systematic review by Molgora et al. [40], especially as many studies support that having high levels of fear of childbirth is associated with a higher frequency of caesarean sections, possibly because fear of childbirth results in a preference for caesarean section [41,42,43].

Untreated FOC is a risk factor for traumatic delivery [44,45], and pregnancy-specific anxiety, including fear of childbirth, which is associated with poor neuro-emotional development in newborns caused by high maternal cortisol levels [46,47].

These findings may be useful as a basis for reflection and for the application of a series of actions in clinical practice, aimed at reducing women’s fear of childbirth. These actions may include encouraging women to attend childbirth preparation classes and providing them with individualised information at each consultation, so that women can also explain and express their fears and feelings. In this way, health professionals can help to reduce women’s fears.

There are several limitations to the present study. One of them could be that the sample was taken from only two hospitals in the south of Spain which are geographically very close to each other. Thus, although the sample size was calculated so as to be representative in number, sampling bias should not be ruled out due to the limited geographical dispersion. Another aspect to mention is that data were self-reported by the participants, which could lead to biases, such as forgetfulness, selective memory, or exaggeration. Participants were also offered a link to the questionnaire that they could fill out from any device with internet access, so having a smartphone or similar was required. This method may have reduced the possibility for researchers to control the sample during the response process. On the other hand, this study also has some strengths. One of them is that validated questionnaires were used for the measures. For the childbirth experience, there was already a validated questionnaire in Spanish (CEQ-E), but the BAS used to measure fear was not validated in our context and had to be validated by cross-cultural adaptation and factor analysis.

## 5. Conclusions

FOC negatively affects the childbirth experience. It is therefore imperative that healthcare providers actively listen to women during the follow-up of pregnancy and encourage their self-expression. This will allow professionals to integrate women’s feelings, experiences, and expectations into subsequent check-ups, during labour, and also in the postpartum period.

Antenatal training is essential for women to be reassured by healthcare professionals by listening to them and giving them hope, fostering feelings of control and confidence, and supporting them to enter the unknown of pregnancy and childbirth. This will increase women’s trust in their healthcare providers, reduce their fear, and enable them to have a more positive childbirth experience.

## Figures and Tables

**Table 1 healthcare-11-00297-t001:** Descriptive analysis of qualitative variables.

Variable	Values	Frequency	Percentage
Marital status	Single	230	55.6
	Married	172	41.5
	Divorced	12	2.8
	Total	414	100
Level of studies	No studies	7	1.8
	Primary studies	112	27.1
	Higher secondary studies or vocational training	168	40.5
	University or postgraduate	127	30.6
	Total	414	100
Employment situation	Unemployed	111	26.8
	Working	232	56
	Household care	71	17.3
	Total	414	100
Economic level (income)	EUR <1000	73	17.6
	Between EUR 1000 and 2000	188	45.4
	Between EUR 2000 and 3000	115	27.8
	EUR >3000	38	9.2
	Total	414	100
Previous miscarriages	Yes	128	31
	No	286	69
	Total	414	100
Attendance to antenatal classes	Yes	179	43.3
	No	235	56.7
	Total	414	100
Mode of onset of labour	Spontaneous	248	59.9
	Induced	133	32
	Planned caesarean	33	8.1
	Total	414	100
Analgesia used during labour	No analgesia	103	25
	Epidural	277	66.9
	Opioids	34	8.1
	Total	414	100
Accompanied during dilation and/or delivery	Yes	356	85.9
	No	58	14.1
	Total	414	100
Type of delivery	Eutocic	255	61.6
	Instrumental	56	13.4
	Caesarean	103	25
	Total	414	100
Stitches after delivery	No stitches	89	21.5
	Episiotomy	74	18
	Tearing	147	35.6
	Caesarean-related	104	25
	Total	414	100
Fear of Childbirth	Low	106	25.7
	Intermediate	229	55.3
	Extreme	79	2
	Total	414	100

**Table 2 healthcare-11-00297-t002:** Correlation between the different quantitative variables.

		Number of Children	Total BAS	Total CEQ-E
Age	Correlation coefficient	0.2 **	−0.31	−0.169 **
	Sig (bilateral)	0.001	0.46	0.001
Number of children	Correlation coefficient		−0.6	0.112 *
	Sig (bilateral)		0.156	0.008
Total BAS	Correlation coefficient			−0.267 **
	Sig (bilateral)			0.001

* The correlation is significant at 0.05 level (bilateral); ** The correlation is significant at 0.001 level (bilateral) (Spearman’s Rho correlation coefficient).

**Table 3 healthcare-11-00297-t003:** Comparison of medians of fear of childbirth (BAS) and childbirth experience (CEQ-E) with sociodemographic variables.

			BAS Test			CEQ-E Test		
		N	Median	Range	*p* Value *	Median	Range	*p* Value *
Level of studies	No studies	7	19	14.22–23.78	0.005	62	48.89–75.11	0.23
	Primary studies	112	16	11.07–20.93		67	56.74–77.26	
	Higher secondary studies or vocational training	168	16	11.09–20.91		65	52.92–77.18	
	University or postgraduate	127	14	9.47–18.53		67	56.36–77.64	
Marital status	Single	230	16	11.32–20.68	0.13	65	53.46–76.54	0.04
	Married	172	15	10.01–19.99		67	56.25–77.75	
	Divorced	12	19	13.87–24.13		63	51.25–74.75	
Economic level (income)	EUR <1000	73	16	10.88–21.12	0.002	61	48.46–73.54	0.001
	Between EUR 1000 and 2000	188	16	11.19–20.81		66	55.43–76.57	
	Between EUR 2000 and 3000	115	16	11.44–20.56		65	53.47–76.53	
	EUR >3000	38	14	9.28–18.72		69	60.33–68.67	

* Kruskal–Wallis test.

**Table 4 healthcare-11-00297-t004:** Comparison of medians of fear of childbirth (BAS) and childbirth experience (CEQ-E) with dichotomous obstetric variables.

			BAS Test			CEQ-E Test		
		N	Median	Range	*p* Value *	Median	Range	*p* Value *
Previous miscarriages	Yes	128	15	10.46–19.54	0.15	67	56.6–77.4	0.001
	No	286	16	11.03–20.97		65	53.43–76.57	
Attendance to antenatal classes	Yes	179	15	10.16–19.84	0.02	65	54.54–75.56	0.21
	No	235	16	11.19–20.81		63	51.16- 74.84	
Accompanied during dilation and/or delivery	Yes	356	16	11.11– 20.89	0.001	67	56.88–77.12	0.001
	No	58	17	12.44–21.56		51	40.5–61.5	

* Mann–Whitney U test.

**Table 5 healthcare-11-00297-t005:** Comparison of medians of fear of childbirth (BAS) and childbirth experience (CEQ-E) with polytomous obstetric variables.

			BAS Test			CEQ-E Test		
		N	Median	Range	*p* Value *	Median	Range	*p* Value *
Mode of onset of labour	Spontaneous	248	15	10.08–19.92	0.09	67	56.65–77.35	0.001
	Induced	133	16	11.54–20.46		64	53.33–74.67	
	Elective caesarean	33	17	11.42–22.58		53	39.78–66.22	
Analgesia used during labour	No analgesia	103	15	10.03–19.97	0.001	71	60.17–81.83	0.001
	Opioids	277	17	12.37–21.63		61	49.92–72.08	
	Epidural	34	16	11.24–20.76		65	53.87–76.13	
Type of delivery	Eutocic	255	15	10.23–19.77	0.08	70	60.4–79.6	0.001
	Instrumental	56	16	11.22–20.78		63	51.56–74.44	
	Caesarean	103	16	11–21		59	47.75–70.25	
Stitches after delivery	No stitches	89	14	8.8–19.2	0.19	70	60.87–79.13	0.001
	Episiotomy	74	16	11.21–20.79		66	55.83–76.17	
	Tearing	147	16	11.36–20.64		67	56.31–77.69	
	Caesarean-related	104	16	11.24–20.76		59	47.73–70.27	

* Kruskal–Wallis test.

**Table 6 healthcare-11-00297-t006:** Binary logistic regression model.

Independent Variables	Sig.	OR	95% C.I. for EXP (B)	
			Lower	Upper
Age	0.001	0.91	0.88	0.95
Income (<1000 euros/month)	0.004	Ref.		
Income (1000–2000 euros/month)	0.002	2.4	1.37	4.23
Income (2000–3000 euros/month)	0.001	2.83	1.52	5.27
Income (>3000 euros/month)	0.002	3.53	1.58	7.89
Miscarriages	0.03	1.58	1.05	2.4
Accompaniment	0.001	3.68	1.91	7.11
Type of delivery (eutocic)	0.001	Ref.		
Type of delivery (instrumental)	0.001	0.41	0.25	0.68
Type of delivery (caesarean)	0.016	0.51	0.3	0.88
Fear of childbirth (FBS score)	0.001	0.9	0.87	0.94

## Data Availability

Data available on request from the authors.

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
