# Peer review of "Antenatal Fear of Childbirth as a Risk Factor for a Bad Childbirth Experience"

_healthcare, 2023, doi:10.3390/healthcare11030297_

Round 1

Reviewer 1 Report

Thank you for the opportunity to review this interesting article on antenatal fear of childbirth as a risk factor for a poor birth experience. This article can be helpful in clinical practice and, above all, for developing future strategies applicable to primary care. However, a number of minor aspects can be revised to improve the overall quality of the article. These are the following:

In the abstract, it would be interesting for the authors to present some numerical results that support their main finding among the most relevant ones.

The authors describe that up to 80% of women may present fear of childbirth. While this is easy to understand because it is logical, it is also striking. I would invite the authors to provide further references to support the significance of this figure.

An interesting finding is the mean age of the women studied, 32 years. In addition, age was negatively correlated with childbirth experience. This aspect is another important finding that would be useful to comment on a bit more in the discussion, as it is relevant.

The findings of this paper are relevant, as they can serve as a basis for developing valuable strategies that reflect on the clinical practice of professionals such as midwives and obstetricians. For this reason, it would be helpful if they proposed some more specific and brief practical ideas that could begin to be developed and, of course, studied in primary care.

The work has some limitations that the authors have described perfectly since. Indeed, the two most important ones are the selection bias and the fact that the questionnaires were self-reported. However, it also has some interesting strengths, such as the sample size or using previously validated scales that have been adapted cross-culturally. I invite the authors to reflect on some of these strengths as well.

Author Response

Thank you for the opportunity to review this interesting article on antenatal fear of childbirth as a risk factor for a poor birth experience. This article can be helpful in clinical practice and, above all, for developing future strategies applicable to primary care. However, a number of minor aspects can be revised to improve the overall quality of the article. These are the following:

In the abstract, it would be interesting for the authors to present some numerical results that support their main finding among the most relevant ones.

Thank you very much for your suggestions. Some important findings have been included in the abstract in numerical form.

The authors describe that up to 80% of women may present fear of childbirth. While this is easy to understand because it is logical, it is also striking. I would invite the authors to provide further references to support the significance of this figure.

Further references are provided to support the fact that a significant proportion of women have fear of childbirth.

An interesting finding is the mean age of the women studied, 32 years. In addition, age was negatively correlated with childbirth experience. This aspect is another important finding that would be useful to comment on a bit more in the discussion, as it is relevant.

The discussion has been expanded by highlighting this interesting finding.

The findings of this paper are relevant, as they can serve as a basis for developing valuable strategies that reflect on the clinical practice of professionals such as midwives and obstetricians. For this reason, it would be helpful if they proposed some more specific and brief practical ideas that could begin to be developed and, of course, studied in primary care.

In the discussion section, more specific and brief ideas are proposed that could be implemented in clinical practice.

The work has some limitations that the authors have described perfectly since. Indeed, the two most important ones are the selection bias and the fact that the questionnaires were self-reported. However, it also has some interesting strengths, such as the sample size or using previously validated scales that have been adapted cross-culturally. I invite the authors to reflect on some of these strengths as well.

Thank you very much. We have tried to reflect the main strengths of the study.

Reviewer 2 Report

Journal: Healthcare

Title: Antenatal fear of childbirth as a risk factor for a bad childbirth 2 experience

Manuscript Number: Healthcare-2137203

KeywordsFear of childbirth; childbirth experience; obstetrics; pregnancy; childbirth

Thank you for the opportunity of reading and reviewing this interesting study. 

Abstract: The abstract is well-written. I would say “childbirth negatively and significantly predicted childbirth experience” instead of correlated.

Introduction: 

Page 2 (line 47): Please, rephrase this sentence to make it clearer “At this end of the spectrum, women may be scrupulous about contraception or resort to voluntary termination of pregnancy triggered by their fear of childbirth.”

Page 2 (line 50): Please, define tokophobia as it is the first time that this term appears in the text.

Please, the introduction is quite short. Please, add two additional paragraphs explaining (paragraph 1) the association between FOC with fear childbirth experience (if there is) and (paragraph 2) with maternal and neonatal health.

In the aims, what do the authors refers to when they say “related consequences”. This should be specified.

Material and Methods:

In the subsection “population and sample” authors refer that the populations consisted on 3269 pregnant women, but this is not clearly the sample size. Please, explain this further. 

What do the authors mean when they say “a total of 3269 pregnant women under follow-up”?

 There is a huge difference between the total sample size of 3269 pregnant women and the 414 women that finally participated. This means that around 2800 women rejected participating. Please, explain this (maybe, but not compulsory, using a participant’s flow diagram.).

Authors explain the “randomization process”, but this is not a randomized controlled trial but a cross-sectional study. This should be avoided and eliminated from this study.

Page 2 (line 92): Please, specify what peripartum pathologies do you refer to.

Why an inclusion criterion was that the newborn was born healthy and without pathologies? If this decrease the birth experience, this should be stated in the introduction.

Was the age of participants taken into consideration to include or exclude participants?

Data Collection: Where were the measures allocated? Google drive? Please, specify.

Data analysis: Please, explain the analysis performed to do a factor analyses.

Results:

Table 1 contains some words in Spanish, please translate into English (for example, “Si” in the previous miscarriages information and accompanied during dilation and/or delivery.

Kruskal-Wallis test was used to compare Medians instead of means. Is this right?

I think the authors have mistranslated “media” for “median”, instead of mean.

Please, explain the results for the factorial analyses performed (for this, move the information provided in the methods section regarding this to the results section). I suggest the authors do (in the future) and additional paper on the psychometric properties of this measure in a Spanish sample.

Discussion:

Is there an association between FOC and/or low birth satisfaction with neonatal outcomes? If so, please, include this information in the discussion.

Please, add a clinical and research implications subsections before the conclusions.

Author Response

Thank you for the opportunity of reading and reviewing this interesting study. 

Abstract: The abstract is well-written. I would say “childbirth negatively and significantly predicted childbirth experience” instead of correlated.

Thank you for your suggestion; it has been changed in the text.

Introduction: 

Page 2 (line 47): Please, rephrase this sentence to make it clearer “At this end of the spectrum, women may be scrupulous about contraception or resort to voluntary termination of pregnancy triggered by their fear of childbirth.”

The sentence has been rewritten to make it clearer.

Page 2 (line 50): Please, define tokophobia as it is the first time that this term appears in the text.

It was defined in the text.

Please, the introduction is quite short. Please, add two additional paragraphs explaining (paragraph 1) the association between FOC with fear childbirth experience (if there is) and (paragraph 2) with maternal and neonatal health.

The introduction was expanded as requested.

In the aims, what do the authors refers to when they say “related consequences”. This should be specified.

The sentence has been rewritten for better understanding.

Material and Methods:

In the subsection “population and sample” authors refer that the populations consisted on 3269 pregnant women, but this is not clearly the sample size. Please, explain this further. 

3269 was the size of the study population. The sample was selected from this population to ensure that it was representative and to obtain a confidence interval of 95% and a margin of error of 5%. Taking these criteria into account and overestimating the number by 20% to allow for possible losses, the sample was set at 414 subjects. This has been explained in detail in lines 98-104.

What do the authors mean when they say “a total of 3269 pregnant women under follow-up”?

This means that a total of 3269 women had their pregnancy under follow-up and monitoring in the study centres. As explained above, the study sample was obtained from this total population.

There is a huge difference between the total sample size of 3269 pregnant women and the 414 women that finally participated. This means that around 2800 women rejected participating. Please, explain this (maybe, but not compulsory, using a participant’s flow diagram.).

3269 was the total population. 414 was the sample that was chosen randomly, systematically, and according to the criteria listed above.

Authors explain the “randomization process”, but this is not a randomized controlled trial but a cross-sectional study. This should be avoided and eliminated from this study.

The sentence has been rewritten so as not to be misleading.

Page 2 (line 92): Please, specify what peripartum pathologies do you refer to.

The requested information was specified in the text.

Why an inclusion criterion was that the newborn was born healthy and without pathologies? If this decrease the birth experience, this should be stated in the introduction.

The fact of having a newborn with certain pathologies increases parents' stress levels and, as a consequence, fear. Above all, the experience of childbirth is affected by these events. Reference was made to this issue in the introduction section.

Was the age of participants taken into consideration to include or exclude participants?

The age of the participants was not considered as they were women who were giving birth and, therefore, women of childbearing age. We assumed the full age range.

Data Collection: Where were the measures allocated? Google drive? Please, specify.

The questionnaires were collected via Google forms. This detail was specified in the text.

Data analysis: Please, explain the analysis performed to do a factor analyses.

The validation process of the questionnaire has been explained in the data analysis section.

Results:

Table 1 contains some words in Spanish, please translate into English (for example, “Si” in the previous miscarriages information and accompanied during dilation and/or delivery.

Thank you for your comment. These words have been translated.

Kruskal-Wallis test was used to compare Medians instead of means. Is this right?

As this variable did not follow a normal distribution, non-parametric tests were used, which actually compare ranges. It is therefore correct to use this test to compare medians.

I think the authors have mistranslated “media” for “median”, instead of mean.

Thank you for your comment, but it is not a translation error. The authors meant to say medians and not means.

Please, explain the results for the factorial analyses performed (for this, move the information provided in the methods section regarding this to the results section). I suggest the authors do (in the future) and additional paper on the psychometric properties of this measure in a Spanish sample.

In the results section, additional information on the factor analysis has been added as requested.

In the future, we will publish the Spanish version of the BAS validated by us.

Discussion:

Is there an association between FOC and/or low birth satisfaction with neonatal outcomes? If so, please, include this information in the discussion.

This information was included in the discussion section.

Please, add a clinical and research implications subsections before the conclusions.

They were added at the end of the discussion section.